# When Self-Supervised Learning Meets Scene Classification: Remote Sensing Scene Classification Based on a Multitask Learning Framework

**Zhicheng Zhao** [1,2] , **Ze Luo** [1,*], **Jian Li** [1], **Can Chen** [1] **and Yingchao Piao** [1]

1    Computer Network Information Center, Chinese Academy of Sciences, Beijing 100190, China;
     zhaozhicheng@cnic.cn (Z.Z.); lijian@cnic.cn (J.L.); chencan@cnic.cn (C.C.); pyc@cnic.cn (Y.P.)
2    University of Chinese Academy of Sciences, Beijing 100049, China
*    Correspondence: luoze@cnic.cn

**Abstract:** In recent years, the development of convolutional neural networks (CNNs) has promoted continuous progress in scene classification of remote sensing images. Compared with natural image datasets, however, the acquisition of remote sensing scene images is more difficult, and consequently the scale of remote sensing image datasets is generally small. In addition, many problems related to small objects and complex backgrounds arise in remote sensing image scenes, presenting great challenges for CNN-based recognition methods. In this article, to improve the feature extraction ability and generalization ability of such models and to enable better use of the information contained in the original remote sensing images, we introduce a multitask learning framework which combines the tasks of self-supervised learning and scene classification. Unlike previous multitask methods, we adopt a new mixup loss strategy to combine the two tasks with dynamic weight. The proposed multitask learning framework empowers a deep neural network to learn more discriminative features without increasing the amounts of parameters. Comprehensive experiments were conducted on four representative remote sensing scene classification datasets. We achieved state-of-the-art performance, with average accuracies of 94.21%, 96.89%, 99.11%, and 98.98% on the NWPU, AID, UC Merced, and WHU-RS19 datasets, respectively. The experimental results and visualizations show that our proposed method can learn more discriminative features and simultaneously encode orientation information while effectively improving the accuracy of remote sensing scene classification.

**Keywords:** self-supervised; multitask; CNN; scene classification; NWPU; deep learning

## 1. Introduction

The aim of remote sensing scene classification is to assign a meaningful land cover type to each patch segmented from a remote sensing image [1–5]. In recent years, with the continuous development of satellite techniques, several remote sensing scene datasets have emerged, and scene classification for remote sensing images has received widespread attention. Compared with natural datasets like ImageNet [6], the acquisition of remote sensing scene images is more difficult, and consequently the scale of the available remote sensing scene datasets is much smaller. Moreover, many problems related to small objects and complex backgrounds arise in remote sensing scenes, presenting serious challenges for classification. As shown in Figure 1, features that contain semantic information may lie within a small area against a complex background. Remote sensing scene classification can play an important role in tasks such as global pollution detection [7,8], land use planning [9], image segmentation [10], object detection [11], and change detection [12]. Therefore, scene classification for remote sensing images has important theoretical research significance as well as important application prospects.

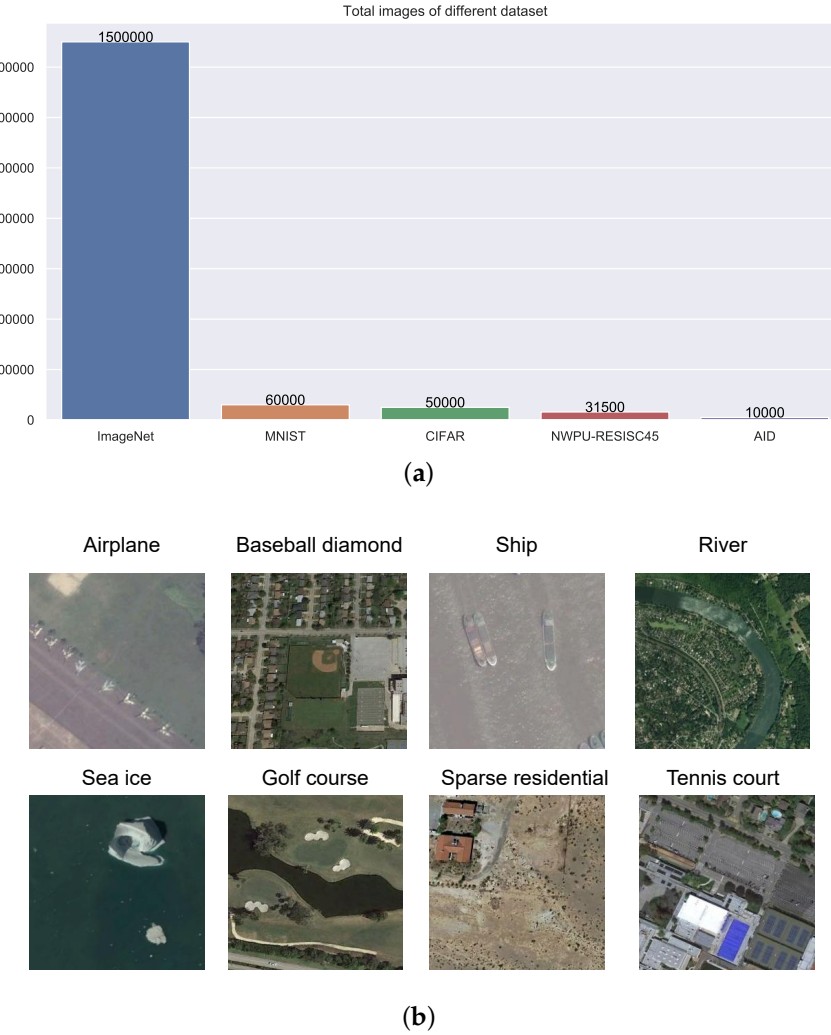

**Figure 1.** Two main problems exist: (**a**) Compared with datasets of natural images, the total numbers of images in remote sensing image datasets are much smaller. (**b**) Compared with natural images, it is difficult to find discriminative semantic features in remote sensing scene images because features containing semantic information may lie within a small area against a complex background.

The methods introduced for remote sensing scene classification in the past two decades can be roughly divided into two types: methods based on manual features and methods based on feature encoding. Manual feature methods include local binary patterns (LBP) [13], histograms of oriented gradients (HOG) [14], and the scale-invariant feature transform (SIFT) [15]. Feature encoding methods include the bag-of-visual-words (BOVW) method [16], the vector of locally aggregated descriptors (VLAD) method [17], and the Fisher vector (FV) method [18]. These methods extract low-level and intermediate-level features of an image. Although these features exhibit rotational invariance and high tolerance to noise, the methods used to extract them require manual parameter adjustment, and their classification accuracy is not sufficiently high.

Recently, deep learning methods have undergone rapid development in the field of computer vision, and several classical convolutional neural network (CNN) models and improved CNN-based network models have emerged. Compared with the previous methods, these models offer stronger feature extraction and generalization abilities [19]. Among them, He et al. [20] proposed the residual network (ResNet) architecture to overcome the difficulty of training a CNN model with many layers and achieved improved classification results on the ImageNet dataset. Huang et al. [21] introduced the dense convolutional network (DenseNet) architecture, in which each layer is connected to every other

layer in a feedforward fashion. Woo et al. [22] proposed the convolutional block attention module (CBAM), a simple and very effective attention module that can be integrated into any feedforward CNN backbone. Experimental results obtained on the ImageNet and CIFAR datasets demonstrate the effectiveness of these methods.

Much progress has been made in the field of remote sensing scene classification based on deep learning methods [23–27]. Wang et al. [28] proposed an improved oriented response network (IORN) model based on oriented response networks (ORNs), which can be used for scene classification for remote sensing images and can extract features with a certain degree of rotational invariance. Inspired by spatial transformation networks, Chen et al. [29] proposed recurrent transformer networks (RTNs), which can learn regional feature representations based on latent relationships. Wang et al. [30] proposed the attention recurrent convolutional network (ArcNet) model, which uses long short-term memory (LSTM) to generate cyclic attention maps and classifies remote sensing scene datasets by weighting these attention maps with high-level CNN-based features. Xue et al. [31] proposed a classification method based on multi-structure deep features fusion(MSDFF). Petrovska [32] used the adoption of transfer learning by fine-tuning pretrained CNNs for end-to-end scene classification. However, although the above methods have achieved good scene classification performance on remote sensing images, they do not make full use of the information contained in the data, and the extracted features are still not sufficiently distinguishable.

In this paper, we introduce a multitask learning framework that combines the tasks of self-supervised learning and classification to enable more efficient use of the original image information and further improve the feature extraction ability of network models. To the best of our knowledge, self-supervised learning has rarely been applied in the field of remote sensing scene classification At the same time, to better combine these two different tasks, we present a new combination mechanism that introduces more randomness to enhance the generalization ability of CNNs. Figure 2 shows a flowchart of the proposed framework. We have conducted extensive tests on current representative remote sensing scene classification datasets and have achieved state-of-the-art results. Our experiments suggest that the combination of these two tasks improves the ability of a CNN model to encode orientation information and helps it learn more discriminative features. The main contributions of this article are as follows:

1.  We propose a multitask learning framework that combines the tasks of self-supervised learning and classification to enhance the generalization ability of CNN models. This framework offers easy model training and can be easily incorporated into other methods.
2.  Different from previous multitask weight adjustment methods, we adopt a dynamic multitask learning weight adjustment strategy called the mixup loss, which not only improves the classification performance but also is not sensitive to the parameter settings.
3.  Comprehensive experiments have been carried out on four remote sensing image datasets to demonstrate the effectiveness of the proposed framework. We have achieved state-of-the-art results on various remote sensing scene classification datasets.

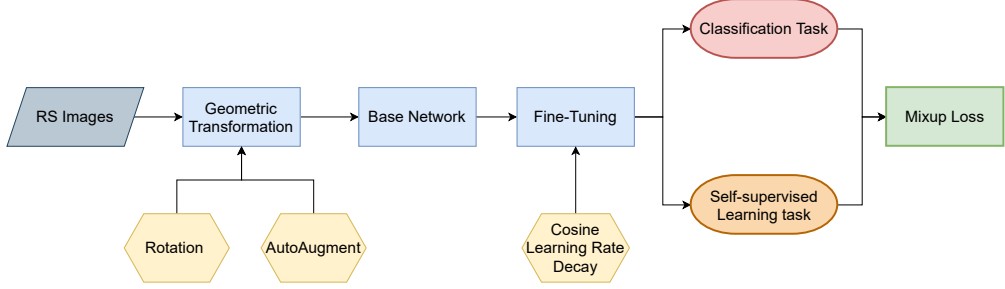

**Figure 2.** Flowchart of the proposed framework.

The remainder of this paper is structured as follows. Section 2 presents related works on modern CNN architectures, multitask learning and self-supervised learning. Section 3 introduces the details of our proposed method. In Sections 4 and 5, the experimental results are reported and discussed. We draw conclusions in Section 6 along with a discussion of future research directions.

## 2. Related Works

In this section, we will briefly review existing related works on modern CNN architectures, self-supervised learning, and multitask learning.

### 2.1. Modern CNN Architectures

Since the development of AlexNet [6], deep CNNs have dominated the task of image classification. Recently, the focus of related research has shifted from engineering handcrafted features to engineering network architectures [21,33,34]. VGG-Net [35] was proposed as a modular network design strategy, in which network blocks of the same type are repeatedly stacked, which simplifies the workflow of network design and transfer learning for downstream applications. ResNet [20] introduced the concept of identity skip connections to alleviate the vanishing gradient problem in deep neural networks and allow networks to learn deeper feature representations. Wide residual networks [36] have been proposed as a novel architecture in which the depth of residual networks is decreased while increasing their width. Building on the success of these pioneering works, Xie et al. [37] proposed a class of deep neural networks for computer vision called ResNeXts based on aggregated residual transformations.

### 2.2. Self-Supervised Learning

Self-supervised learning is a general learning framework that relies on pretext tasks that can be formulated using only unsupervised data [38,39]. It is a new paradigm that lies between unsupervised and supervised learning. It can reduce the need for large amounts of annotated data, which can be challenging to obtain these annotated data. A pretext task is designed such that solving it will require the learning of a useful image representation. For example, patch-based methods [40–42] predict the relative locations of multiple randomly sampled image patches. In addition to patch-based methods, there are self-supervised techniques that employ image-level losses. Zhang et al. [43] proposed grayscale image colorization as a pretext task. The authors of [44] designed a pretext task that involves predicting the angle of a rotation transformation that has been applied to an input image.

### 2.3. Multitask Learning

Multitask learning (MTL) is a learning paradigm in machine learning with the aim of leveraging useful information inferred from multiple related tasks to help improve the generalization performance for all tasks [45,46]. MTL improves generalization by leveraging the domain-specific information contained in the training signals for related tasks. This is achieved by training a model for all tasks in parallel while using a shared representation. Many deep MTL methods [47–49] assume that the first several hidden layers are shared among the different tasks, while the subsequent layers contain task-specific parameters. The powerful representation capabilities of deep networks provide increased space for deep MTL.

## 3. Methods

In this section, we will introduce the implementations and details of the proposed MTL framework for remote sensing scene classification. Training for the primary task is performed based on ground-truth labels, whereas training for the auxiliary task is performed based on geometric transformation labels.

### 3.1. Self-supervised Learning Task

Recent self-supervised learning studies have shown that high-level semantic representations can be learned by predicting labels that can be obtained from the input signals without any human annotation [38,50,51]. Intuitively, a good CNN model should learn to recognize the orientations of different scenes in remote sensing scene images. In the framework proposed in this paper, we implement self-supervised learning by producing four copies of a single remote sensing image by rotating it by 0, 90, 180, and 270 degrees and using a single network to predict the rotation angle as a 4-class classification task. As shown in Figure 3, the basic idea behind using these image rotations as the set of geometric transformations is founded on the simple fact that it is essentially impossible for a CNN model to effectively perform the above rotation recognition task unless it has first learned to recognize and detect classes of objects as well as their semantic features in images.

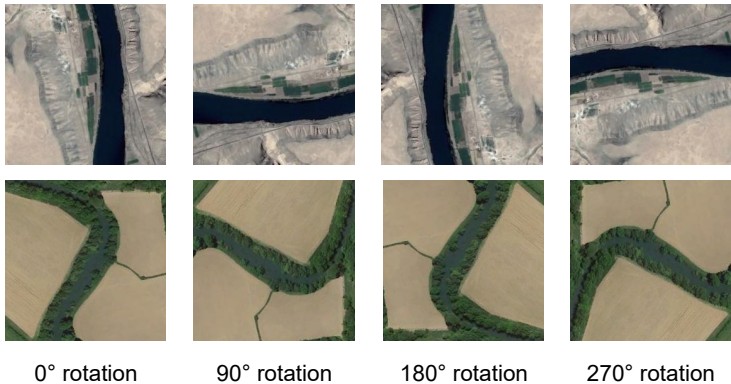

<div align="center">0° rotation    90° rotation    180° rotation    270° rotation</div>

**Figure 3.** Example images rotated by random multiples of 90 degrees (e.g., 0, 90, 180, or 270 degrees). The core intuition of self-supervised learning is that if the CNN model is not aware of the concepts of the objects depicted in the images, it cannot recognize the rotation that was applied to them.

The purpose of the whole self-supervised learning task is to train a CNN model $F(.)$ to estimate the geometric transformation applied to an image. Specifically, we first define $G$ as a set of $K$ discrete geometric transformations:

$$G = \{g(\cdot|y)\}_{y=1}^{K} \tag{1}$$

where $g(\cdot|y)$ is the operator that is applied to image X and the geometric transformation with label $y$ yields the transformed image $X^y = g(X|y)$.

The CNN model $F(.)$ takes as input an image $X^{y*}$ (where the label $y^*$ is unknown to the model) and yields as output a feature descriptor over all possible geometric transformations:

$$F(X^{y*}|\theta) = \{F^y(X^{y*}|\theta)\}_{y=1}^{K} \tag{2}$$

where $F^y(X^{y*}|\theta)$ is the feature descriptor for the geometric transformation with label $y$ and $\theta$ represents the learnable parameters of model $F(.)$.

Therefore, given a set of $N$ training images $X = \{X_i\}_{i=0}^{N}$, as an auxiliary task, the loss function $L_A$ for the self-supervised learning task is calculated as follows,

$$f = F_{avg}(F(X^{y*}|\theta)) \tag{3}$$

$$\hat{p} = \text{softmax}(W_1 f + B_1) \tag{4}$$

$$loss(X_i, \theta) = -\frac{1}{K} \sum_{i}^{K} y_i log(\hat{p}_i) \tag{5}$$

$$L_A(X, \theta, W_1, B_1) = \frac{1}{N} \sum_{i=i}^{N} loss_i \tag{6}$$

where $F_{avg}$ denotes the global average pooling (GAP) operator, $f$ is a feature vector learned after the CNN model $F$ and the GAP operator (e.g., for ResNet and ResNeXt, $f$ is a 2048-dimensional feature vector), $W_1$ denotes the weights of the final layer, and $B_1$ refers to the corresponding bias.

As shown in Figure 4, in this paper, we define the set of geometric transformations $G$ as all image rotations by multiples of 90 degrees, i.e., 2D image rotations by 0, 90, 180, and 270 degrees. More formally, if $\text{Rotate}(X, \phi)$ is an operator that rotates image $X$ by $\phi$ degrees, then our set of geometric transformations consists of the $K = 4$ image rotations $G = \{g(X|y)\}_{y=1}^{4}$, where $g(X|y) = \text{Rotate}(X, (y-1)90)$.

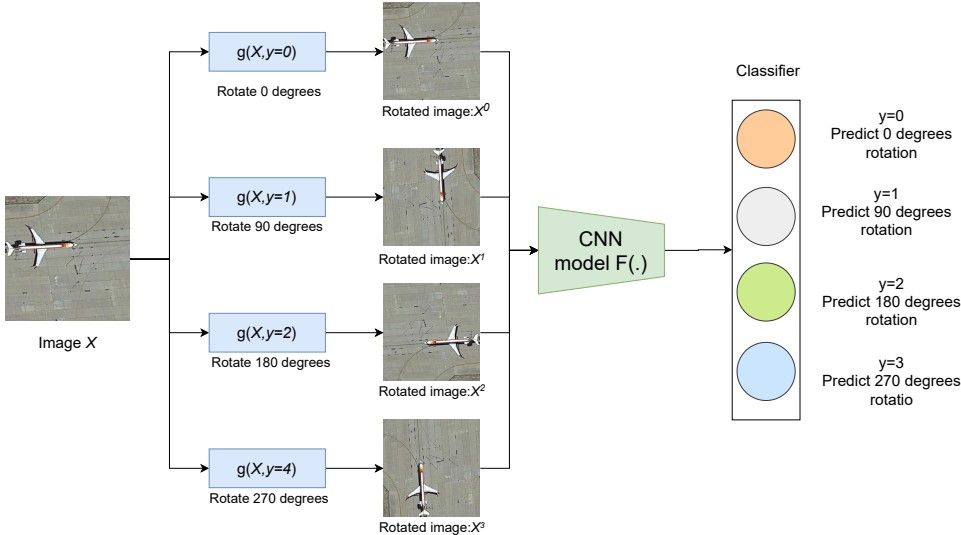

**Figure 4.** Illustration of the self-supervised learning task that we propose for semantic feature learning. Given four possible geometric transformations, i.e., rotations by 0, 90, 180, and 270 degrees, we train a CNN model $F(.)$ to recognize the rotation that has been applied to the image that it receives as input.

### 3.2. Classification Task

We use the cross-entropy loss as the classification loss for label prediction. For the classification loss $L_P$,

$$\tilde{p} = \text{softmax}(W_2 f + B_2) \tag{7}$$

$$L_P(X, \theta, W_2, B_2) = -\frac{1}{N} \sum_{i=i}^{N} y_i log(\tilde{p}_i) \tag{8}$$

where $f$ is a 2048-dimensional feature vector, $W_2$ denotes the weights of the final layer, and $B_2$ refers to the corresponding bias.

### 3.3. Combination of the Two Tasks

Combining the self-supervised learning and classification tasks can help a baseline CNN model improve its ability to encode orientation information and speed up optimization during training. The common approach for utilizing self-supervised labels for another task is to optimize the losses for the two tasks considering a shared feature space; that is, a model is trained for both tasks in the MTL framework [38,52,53]. Thus, in a fully supervised setting, one can formulate the multitask objective with self-supervision as follows,

$$L_{MT}(X, \theta, W_1, B_1, W_2, B_2) = L_P + L_A \tag{9}$$

where $L_P$ denotes the loss for the classification task and $L_A$ denotes the loss for the self-supervised learning task. The above loss also forces the primary classifier $\sigma(f(\cdot;\theta);u)$ to be invariant with respect to the transformations $\{t_j\}$. Thus, for the aforementioned reason, the usage of additional self-supervised labels does not guarantee performance improvement, especially in the fully supervised setting.

Another common approach for combining two tasks is to specify two fixed parameters: $\lambda_1$ and $\lambda_2$. However, determining the specific values of these parameters is very challenging. In some cases, if the appropriate parameters cannot be determined, the classification performance may even decrease.

$$L_{MT}(X, \theta, W_1, B_1, W_2, B_2) = \lambda_1 * L_P + \lambda_2 * L_A \tag{10}$$

Motivated by these issues and inspired by methods of data augmentation [54,55], we introduce a simple and useful combination strategy called the mixup loss. This method does not require the determination of parameter values, can introduce more randomness into the network model, and can improve the feature representation ability of the model.

$$L_{MT}(X, \theta, W_1, B_1, W_2, B_2) = \lambda * L_P + (1 - \lambda) * L_A \tag{11}$$

where the parameter $\lambda$ is a random float number from 0 to 1. And it is generated from the $Beta(\alpha, \alpha)$ distribution for $\alpha \in (0, \infty)$.

In probability theory and statistics, the beta distribution is a family of continuous probability distributions defined on the interval [0, 1] parameterized by two positive shape parameters, denoted by $\alpha$ and $\beta$, that appear as exponents of the random variable and control the shape of the distribution. The beta distribution has been applied to model the behavior of random variables limited to intervals of finite length in a wide variety of disciplines. To simplify the setting, we set $\alpha = \beta$ in this paper so that we need to set only $\alpha$. The corresponding probability densities for different values of the parameter $\alpha$ are shown in Figure 5. Specifically, when $\alpha = 1$, the $Beta(1, 1)$ distribution is equivalent to a uniform distribution.

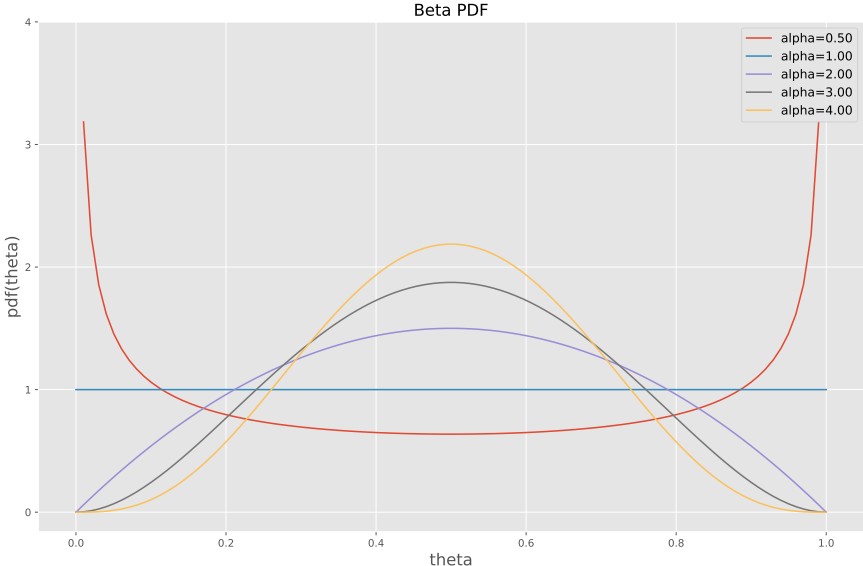

**Figure 5.** The probability density functions for the beta distribution with different values of $\alpha$.

### 3.4. MTL Framework

The MTL framework is illustrated in Figure 6. The inputs to the proposed method are obtained from images $X \in R^{C \times H \times W}$. Then, the inputs $X$ are geometrically transformed using the $K = 4$ image rotations $G = \{g(X|y)\}_{y=1}^4$, where $g(X|y) = \text{Rot}(X, (y-1)90)$. After geometric transformation,

the inputs have become $X \in R^{4C \times H \times W}$. Then, the inputs are fed into the backbone $F$, through the GAP operator [56], to obtain the feature description $f$, which has different sizes for different backbones, e.g., for ResNet, $f$ is a 2048-dimensional feature vector.

The network is trained on two tasks. The primary task is the classification task, of which the aim is to identify a determinate category for each remote sensing scene. The auxiliary task is a self-supervised learning task in which the aim is to predict the rotation label. The cross-entropy losses for both tasks are combined using the mixup loss strategy. Finally, in our MTL framework, a model is trained to minimize the two losses jointly. This method of combination forces the model to learn a discriminative feature representation with good rotational invariance and robustness.

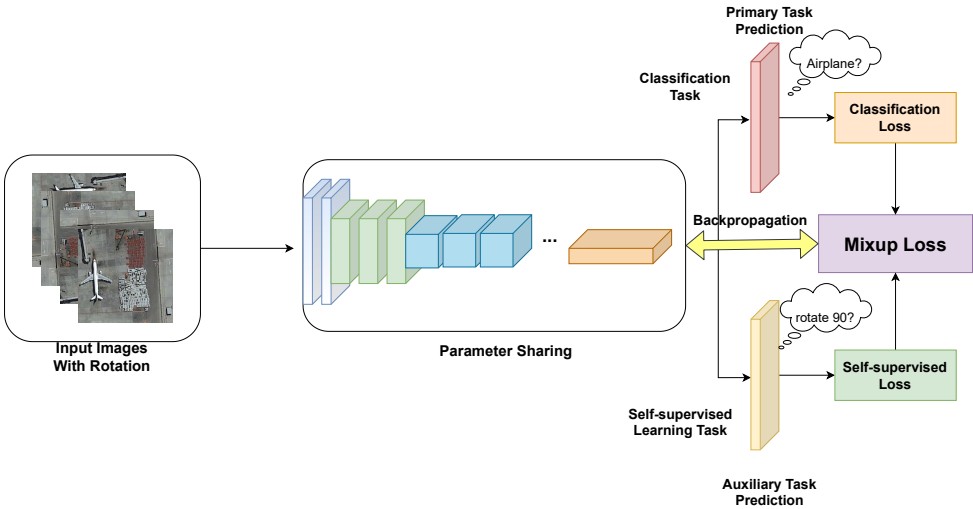

**Figure 6.** Architecture of the proposed multitask learning (MTL) framework. Multiple input images are generated from a single image by rotating by 90, 180, and 270 degrees. The network is trained on two tasks. The main task is the classification task, of which the aim is to identify a determinate category for each remote sensing scene. The auxiliary task is a self-supervised learning task in which the aim is to obtain the rotation label. The two tasks are combined using the mixup loss strategy.

Regarding the backbone, three representative CNN architectures (VGG, ResNet, and ResNeXt) that have been fully trained on the ImageNet dataset are chosen as feature map extractors, considering their popularity in the field of remote sensing scene classification. If the input image size is $256 \times 256$ pixels, the output feature maps of VGG, ResNet, and ResNeXt have dimensions of $8 \times 8 \times 512$, $8 \times 8 \times 2048$, and $8 \times 8 \times 2048$, respectively. The different building blocks of these three CNN architectures are shown in Figure 7. The influence of the three pretrained CNN backbones on the classification results is discussed in Section 4.3. In addition, brief introductions to these models follow.

- VGG: Simonyan et al. [35] presented a thorough evaluation of networks of increasing depth using an architecture with very small ($3 \times 3$) convolutional filters, and the results showed that a significant improvement over prior state-of-the-art configurations could be achieved by increasing the depth to 16–19 convolutional weight layers. The most common network configuration used in remote sensing scene classification is VGG-16 (containing 13 convolutional layers and three fully connected layers).
- ResNet: Deeper neural networks are more difficult to train [20]. To solve the problem of network degradation caused by an increase in depth, the layers of deep ResNets are reformulated to learn residual functions with reference to the layer inputs. The residual learning framework can ease the training of networks that are substantially deeper than those used previously. ResNet-50 and ResNet-101 are widely used as backbones in many tasks.
- ResNeXt: Based on ResNet [20] and Inception [19], Xie et al. [37] introduced a new hyperparameter called the cardinality (the size of the set of transformations) as an essential factor in addition to the

dimensions of depth and width. These authors empirically showed that even under the restricted condition of maintaining the model complexity, it is possible to improve the classification accuracy of a model by increasing the cardinality.

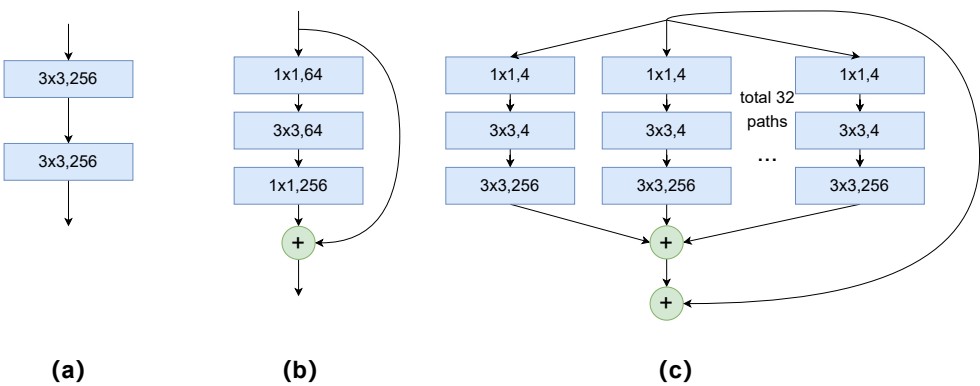

**Figure 7.** Three different network building blocks: (**a**) A VGG block. (**b**) A ResNet block. (**c**) A ResNeXt block.

### 3.5. Aggregation of Information

During the test phase, inspired by ensemble learning [57,58], we rotate a single test image by 90, 180, and 270 degrees. By means of the CNN model and the GAP operator, we can then obtain four feature maps, $f_1$, $f_2$, $f_3$, and $f_4$. Intuitively, we can aggregate the different information contained in these feature maps by taking the mean of the four descriptors as follows (see Figure 8),

$$f_{mean} = (f_1 + f_2 + f_3 + f_4)/4 \tag{12}$$

where $f_1, f2, f_3, f_4, f_{mean} \in R^C$ and $f_1$, $f_2$, $f_3$, and $f_4$ are generated from image $X$ by rotating it by 0, 90, 180, and 270 degrees.

Compared with single-image prediction, our experiments show that by aggregating the feature descriptors in this way, a gain in accuracy can be achieved, thus indicating the effectiveness of such aggregation.

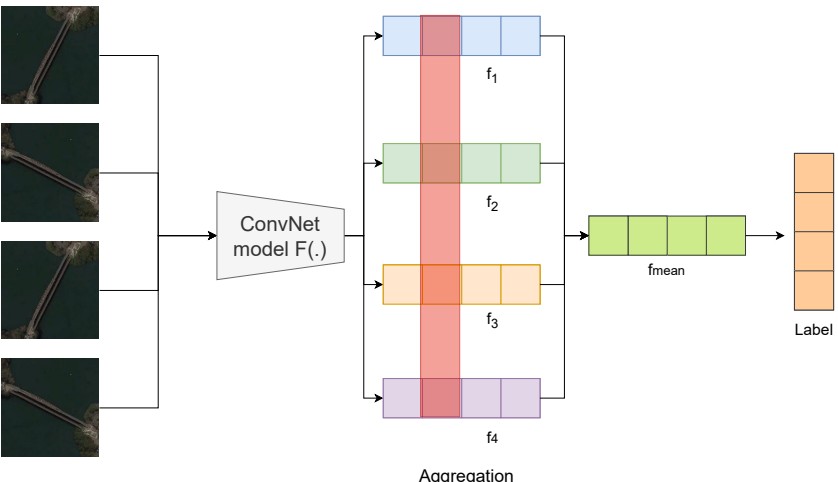

**Figure 8.** In the test stage, $f_{mean}$ is obtained by aggregating the four different feature descriptors $f_1$, $f2$, $f_3$, and $f_4$. Then, $f_{mean}$ is used for prediction.

## 4. Experimental Results

*4.1. Datasets*

To prove the effectiveness of the framework proposed in this paper, experiments carried out on four datasets commonly used in remote sensing scene classification are reported. Table 1 shows the details of the four datasets. Figure 9 presents example images from the different datasets.

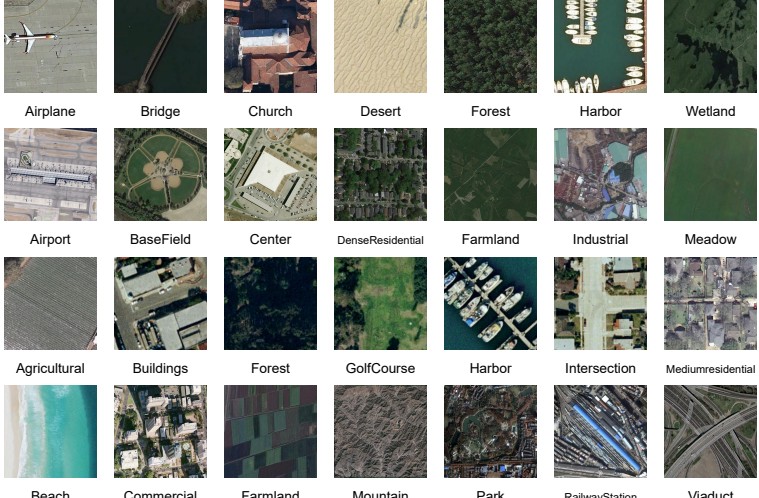

**Figure 9.** Example images from the four remote sensing scene classification datasets. From the top row to the bottom row: NWPU-RESISC45, AID, UC Merced, and WHU-RS19.

The NWPU-RESISC45 dataset [4] is currently the largest publicly available benchmark dataset for remote sensing scene classification. It contains 45 classes of scene images. Each class contains 700 images with dimensions of $256 \times 256$ pixels, and the spatial resolution of the images varies from approximately 0.2 to 30 m. From each class, images were randomly selected at ratios of 10:90 and 20:80 to obtain the training and test sets.

The Aerial Image Dataset (AID) [59] contains 30 classes of scene images; each class contains approximately 200 to 400 samples, for a total of 10,000 images, and each image is $600 \times 600$ pixels in size. From each class, images were randomly selected at ratios of 20:80 and 50:50 to obtain the training and test sets.

The UC Merced land use dataset [60] is composed of 2100 overhead scene images divided into 21 land use scene classes. Each class consists of 100 aerial images measuring $256 \times 256$ pixels, with a spatial resolution of 0.3 m per pixel in the red-green-blue color space. To date, this dataset has been very popular and has been widely used for scene classification and retrieval tasks on remote sensing images.

The WHU-RS19 dataset [61] contains 19 classes of scene images, each containing approximately 50 samples, for a total of 1005 images, and each image is $600 \times 600$ pixels in size. This dataset has also been widely adopted to evaluate various scene classification methods.

**Table 1.** Comparison of the four different remote sensing scene datasets.

| Dataset | Images per Class | Scene Classes | Total |
|---|---|---|---|
| NWPU-RESISC45 | 700 | 45 | 31,500 |
| AID | 200–480 | 30 | 10,000 |
| UC Merced | 100 | 21 | 2100 |
| WHU-RS19 | 50 | 19 | 1005 |

*4.2. Implementation Details*

We tested our method on the four datasets. The backbone networks, including VGG-16, ResNet-50, ResNet-101, ResNeXt-50, and ResNeXt-101, were pretrained on ImageNet and then fine-tuned on the different datasets. We implemented our proposed architecture with the MXNet framework. We resized all the images to $256 \times 256$ pixels using the Nesterov accelerated gradient (NAG) optimization method with a learning rate of 0.005. The learning rates were adjusted in accordance with a cosine schedule [21,54]. The experiments were implemented on a workstation with two 2.2 GHz ten-core CPUs and 64 GB of memory. Training under our MTL framework was implemented with two NVIDIA RTX Titan GPUs for acceleration. To ensure fair comparisons, all networks were trained for 100 epochs. It should be noted that in the last 20 epochs, we trained the networks only on the classification task for better convergence. To obtain reliable results on all four datasets, we repeated the experiment 5 times for each training ratio with randomly selected training samples; the means and standard deviations of the results are reported. Considering the results on all four datasets, we set $\alpha$ to 1 for all subsequent experiments.

In addition, we used the currently popular data augmentation strategy called AutoAugment [62]. AutoAugment is a strategy for augmenting training data with transformed images in which the transformations are learned adaptively. Sixteen different types of image jittering transformations are introduced, and from these, one augments the data based on 24 different combinations of two consecutive transformations, such as shift and color jittering. In our experiments, we used the AutoAugment strategy trained on ImageNet.

*4.3. Ablation Study*

To validate the effectiveness of our proposed framework, we conducted ablation experiments on the four datasets. Table 2 presents the results of the ablation study of models under two settings:

- Base Network: As the baselines for these experiments, we adopted three representative backbone architectures: VGG, ResNet, and ResNeXt. For VGG, we chose VGG-16 as the backbone. For ResNet, we chose ResNet-50 and ResNet-101 as the backbones. For ResNeXt, we chose ResNeXt-50 and ResNeXt-101 as the backbones.
- Base Network+MTL: We fine-tuned the backbones with our proposed MTL framework.

**Table 2.** Comparison of model performance with and without MTL. The bold results are obtained by our proposed method.

| Method | NWPU (0.2) | AID (0.5) | UC Merced (0.8) | WHU-RS19 (0.6) |
|---|---|---|---|---|
| VGG-16 | $90.36 \pm 0.18$ | $94.64 \pm 0.51$ | $97.14 \pm 0.56$ | $97.06 \pm 0.48$ |
| **VGG-16+MTL (ours)** | $91.50 \pm 0.27$ | $94.78 \pm 0.43$ | $98.29 \pm 0.44$ | $98.37 \pm 0.39$ |
| ResNet-50 | $91.86 \pm 0.19$ | $95.96 \pm 0.17$ | $98.69 \pm 0.49$ | $98.61 \pm 0.41$ |
| **ResNet-50+MTL (ours)** | $\mathbf{92.71 \pm 0.12}$ | $\mathbf{96.22 \pm 0.13}$ | $\mathbf{98.78 \pm 0.38}$ | $\mathbf{98.73 \pm 0.33}$ |
| ResNet-101 | $92.52 \pm 0.17$ | $96.34 \pm 0.22$ | $98.91 \pm 0.27$ | $98.92 \pm 0.36$ |
| **ResNet-101+MTL (ours)** | $\mathbf{93.93 \pm 0.16}$ | $\mathbf{96.61 \pm 0.19}$ | $\mathbf{98.91 \pm 0.49}$ | $\mathbf{99.06 \pm 0.31}$ |
| ResNeXt-50 | $92.66 \pm 0.14$ | $96.29 \pm 0.31$ | $98.73 \pm 0.46$ | $98.75 \pm 0.41$ |
| **ResNeXt-50+MTL (ours)** | $\mathbf{93.83 \pm 0.21}$ | $\mathbf{96.55 \pm 0.14}$ | $\mathbf{99.02 \pm 0.35}$ | $\mathbf{99.13 \pm 0.39}$ |
| ResNeXt-101 | $93.68 \pm 0.31$ | $96.52 \pm 0.23$ | $98.96 \pm 0.31$ | $98.88 \pm 0.36$ |
| **ResNeXt-101+MTL (ours)** | $\mathbf{94.21 \pm 0.15}$ | $\mathbf{96.89 \pm 0.18}$ | $\mathbf{99.11 \pm 0.25}$ | $\mathbf{98.98 \pm 0.26}$ |

From the results, we can see that as the model becomes increasingly complex, the classification results improve. Our MTL framework enables performance improvements of the three different backbones on the four different datasets. In addition to these performance improvements, we can also see that our method results in small standard deviations, indicating that models trained using our framework are generally more stable and robust than the base networks.

*4.4. Evaluation of Aggregation Prediction*

In Table 3, we compared aggregation prediction with single-image prediction (see Section 3.5 for details). It should be noted that these experiments were conducted on the NWPU-RESISC45 dataset under training ration of 20%. As can be seen from the results, using aggregation of prediction can perform better results than singe-image prediction. This conforms that this way of aggregation can make more effective use of the original input images.

**Table 3.** Comparisons of aggregation prediction and single-image prediction. The bold results are obtained by our proposed method.

| Methods | Prediction Methods | Accuracy(%) |
|---|---|---|
| ResNext-50+MTL | single-image | $93.52 \pm 0.29$ |
| | aggregation | **$93.83 \pm 0.21$** |
| ResNext-101+MTL | single-image | $93.92 \pm 0.25$ |
| | aggregation | **$94.21 \pm 0.15$** |

*4.5. Results on Different Datasets*

We conduct experiments on four representative remote sensing scene classification datasets including NWPU, AID, UC Merced, and WHU-RS19. Tables 4–7 show the results obtained on the four datasets. We compare our method with several state-of-the-art methods on these datasets. For WHU-RS19, as there are fewer methods conducted on this dataset, the methods for comparison are different from the other three datasets. Note that the relevant results are referred to the original papers.

4.5.1. Results on NWPU-RESISC45

Table 4 compares the classification performance of CNNs trained under our MTL framework and existing state-of-the-art methods on the highly challenging NWPU-RESISC45 dataset with training proportions of 10% and 20%. This dataset is more challenging because the model needs to predict labels of many testing data by utilizing few training samples. We show the classification results produced by some state-of-the-art methods such as Recurrent Transformer Network (RTN) [29] and Multi-Granularity Canonical Appearance Pooling (MG-CAP) [63]. It can be observed that the combination of ResNet-101 and MTL yields a top-1 accuracy of 94.21%, representing state-of-the-art performance compared with other methods. The good performance of the proposed method further verifies the effectiveness of combining self-supervised learning with pretrained CNN models.

**Table 4.** Results of our proposed method and other methods considered for comparison in terms of overall accuracy (%) and standard deviation (%) on the NWPU-RESISC45 dataset for training proportions of 10% and 20%. The bold results are obtained by our proposed method.

| Method | Training Proportion | |
|---|---|---|
| | 10% | 20% |
| GoogLeNet+SVM | $82.57 \pm 0.12$ | $86.02 \pm 0.18$ |
| D-CNN with GoogLeNet [64] | $86.89 \pm 0.10$ | $90.49 \pm 0.15$ |
| RTN [29] | 89.90 | 92.71 |
| MG-CAP (Log-E) [63] | $89.42 \pm 0.19$ | $91.72 \pm 0.16$ |
| MG-CAP (Bilinear) [63] | $89.42 \pm 0.19$ | $91.72 \pm 0.16$ |
| MG-CAP (Sqrt-E) [63] | $90.83 \pm 0.12$ | $92.95 \pm 0.11$ |
| ResNet-101 | $89.41 \pm 0.16$ | $92.51 \pm 0.17$ |
| **ResNet-101+MTL (ours)** | **$91.61 \pm 0.22$** | **$93.93 \pm 0.16$** |
| ResNeXt-101 | $91.18 \pm 0.29$ | $93.68 \pm 0.31$ |
| **ResNeXt-101+MTL (ours)** | **$91.91 \pm 0.18$** | **$94.21 \pm 0.15$** |

Figure 10 shows the confusion matrix generated from the best classification results obtained by ResNeXt-101+MTL with a training proportion of 20%. As seen from the confusion matrix, classification accuracies greater or equal to 90% are achieved for 38 of the 45 categories, with the accuracy for the "cloud" category being 100%. The greatest confusion is observed between the "palace" and "church" categories; thus, we infer that scenes in these categories possess similar features.

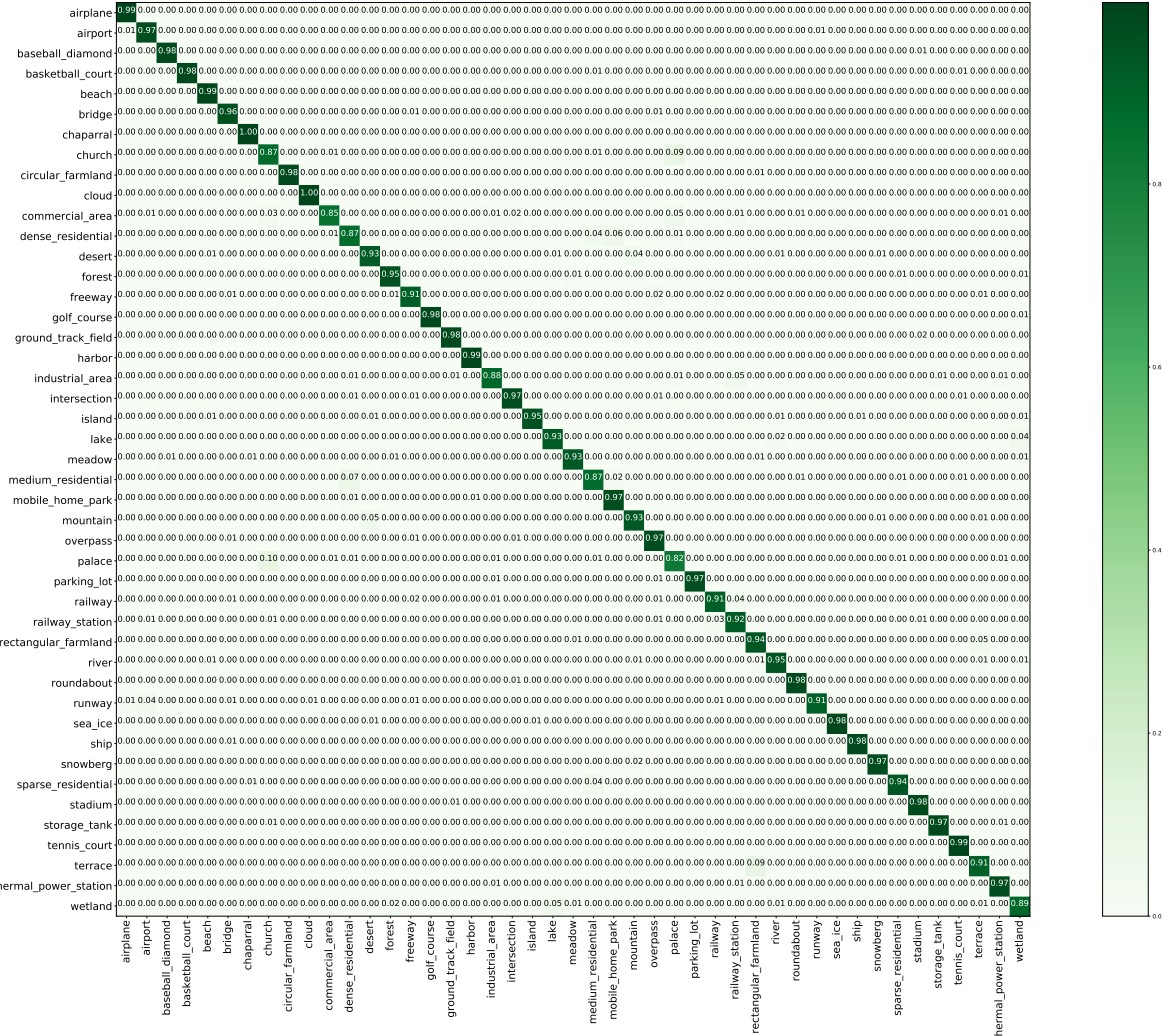

**Figure 10.** Confusion matrix of the proposed method on the NWPU dataset with a training proportion of 20%.

### 4.5.2. Results on AID

Our proposed framework was also tested on AID to demonstrate its effectiveness compared with other state-of-the-art methods on the same dataset. The results are shown in Table 5. It can be seen that the combination of the self-supervised learning and classification tasks again results in the best performance, with accuracies of 96.89% and 93.96% resulting from training using 50% and 20% of the samples, respectively.

As seen from an analysis of the confusion matrix, shown in Figure 11, classification accuracies greater or equal to 90% are achieved for 28 of the 30 categories, with the accuracies for the "baseballfield", "bridge", "forest", "meadow", "pond", and "viaduct" classes being 100%. These findings indicate that the MTL framework enables the model to learn the differences in spatial information among these scene classes with the same image distribution and effectively distinguish them. Meanwhile, the "school" class is easily confused with the 'commercial' class because they have

the same image distribution. In addition, the "resort" class is commonly misclassified as 'park' due to the presence of certain similar objects, such as green belts and ponds.

**Table 5.** Results of our proposed method and other methods considered for comparison in terms of overall accuracy and standard deviation (%) on AID. The bold results are obtained by our proposed method.

| Method | Training Proportion | |
| --- | --- | --- |
| | 20% | 50% |
| GoogLeNet+SVM | 87.51 ± 0.11 | 95.27 ± 0.10 |
| D-CNN with GoogLeNet [64] | 86.89 ± 0.10 | 90.49 ± 0.15 |
| RTN [29] | 92.44 | - |
| MG-CAP (Log-E) [63] | 90.17 ± 0.19 | 94.85 ± 0.16 |
| MG-CAP (Sqrt-E) [63] | 90.83 ± 0.12 | 92.95 ± 0.11 |
| MG-CAP (Bilinear) [63] | 92.11 ± 0.15 | 95.14 ± 0.12 |
| MG-CAP (Sqrt-E) [63] | 93.34 ± 0.18 | 96.12 ± 0.12 |
| ResNet-101 | 93.31 ± 0.19 | 96.34 ± 0.22 |
| **ResNet-101+MTL (ours)** | **93.67 ± 0.21** | **96.61 ± 0.19** |
| ResNeXt-101 | 93.11 ± 0.22 | 96.52 ± 0.23 |
| **ResNeXt-101+MTL (ours)** | **93.96 ± 0.11** | **96.89 ± 0.18** |

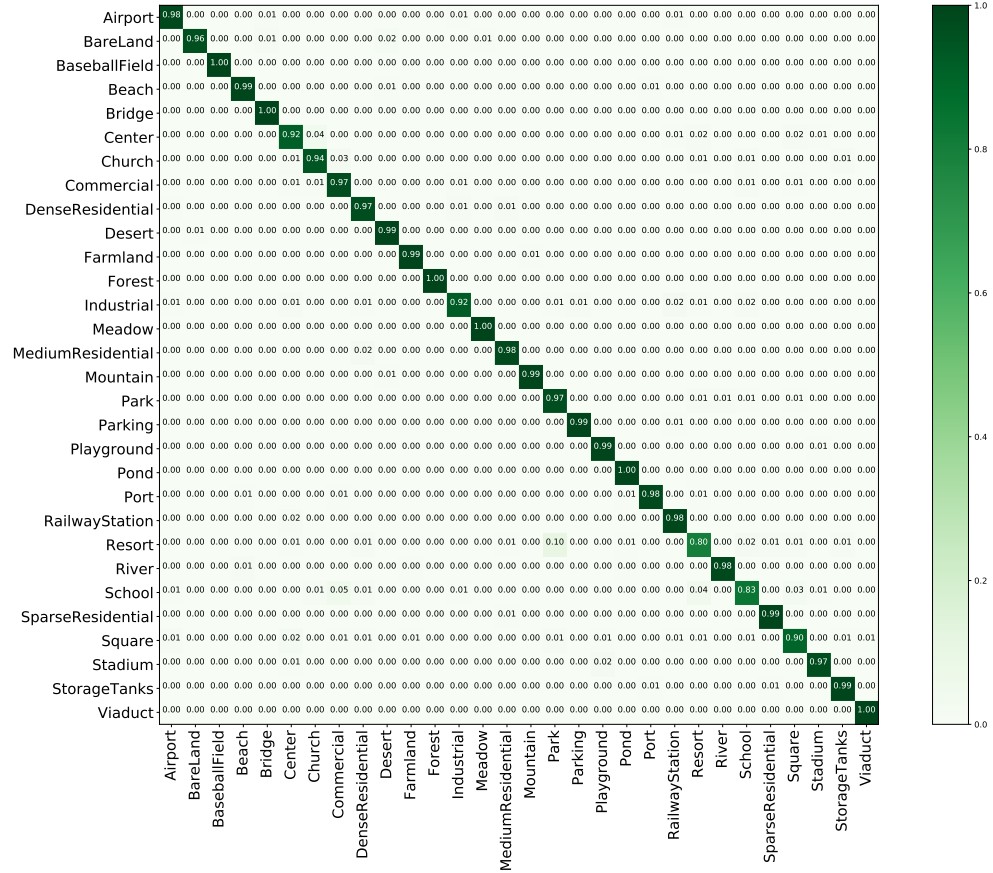

**Figure 11.** Confusion matrix of the proposed method on AID with a training proportion of 50%.

### 4.5.3. Results on UC Merced

To further evaluate the classification performance of the proposed method, a comparative evaluation against several state-of-the-art classification methods on the UC Merced land use dataset is shown in Table 6. We can see that due to the relative lack of image variations and diversity in

this dataset, the overall accuracy is almost saturated. In addition, due to the limited dataset scale, the standard deviations are larger than those on NWPU and AID.

**Table 6.** Results of our proposed method and other methods considered for comparison in terms of overall accuracy and standard deviation (%) on the UC Merced dataset (training proportion of 80%). The bold results are obtained by our proposed method.

| Method | Accuracy |
|---|---|
| GoogLeNet+SVM | 96.82 ± 0.20 |
| D-CNN with GoogLeNet [64] | 97.07 ± 0.12 |
| RTN [29] | 98.60 ± 0.26 |
| MG-CAP (Log-E) [63] | 98.45 ± 0.12 |
| MG-CAP (Bilinear) [63] | 98.60 ± 0.26 |
| MG-CAP (Sqrt-E) [63] | 99.0 ± 0.10 |
| ResNet-101 | 98.91 ± 0.27 |
| **ResNet-101+MTL (ours)** | **99.02 ± 0.35** |
| ResNeXt-101 | 98.96 ± 0.31 |
| **ResNeXt-101+MTL (ours)** | **99.11 ± 0.25** |

Figure 12 shows the confusion matrix generated from the best classification results obtained by ResNeXt+MTL with a training proportion of 80%. As shown, accuracies greater or equal to 90% are achieved for all 21 categories, with the majority showing accuracies of 100%. Indeed, an accuracy as low as 90% is seen only for the "dense residential" and "medium residential" classes, which can be easily confused with each other. We infer that it is difficult to distinguish these classes because of their similar building structures and densities.

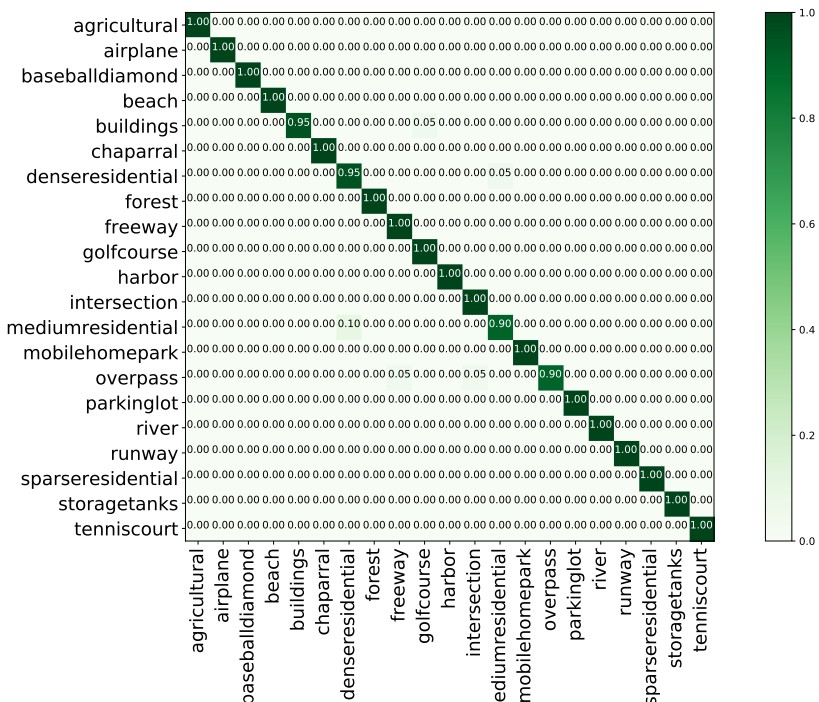

**Figure 12.** Confusion matrix of the proposed method on the UC Merced land use dataset with a training proportion of 80%.

### 4.5.4. Results on WHU-RS19

Finally, to validate the performance of the proposed method on a small dataset, we conducted experiments on the WHU-RS19 dataset, which has the smallest scale among the four datasets. Due to

the very few available training samples, the accuracy of different network models tends to be saturated. Nevertheless, compared with advanced ensemble learning methods, our method still achieves a slight improvement.

**Table 7.** Results of our proposed method and other methods considered for comparison in terms of overall accuracy and standard deviation (%) on the WHU-RS19 dataset (training proportion of 60%). The bold results are obtained by our proposed method.

| Method | Accuracy |
|---|---|
| DCA by concatenation [65] | 98.70 ± 0.23 |
| Fusion by addition [65] | 98.65 ± 0.43 |
| ResNet-101 | 98.62 ± 0.27 |
| **ResNet-101+MTL (ours)** | **98.96 ± 0.31** |
| ResNeXt-101 | 98.88 ± 0.36 |
| **ResNeXt-101+MTL (ours)** | **98.98 ± 0.26** |

Figure 13 shows the confusion matrix generated from the best classification results obtained by ResNeXt+MTL with a training ratio of 60%. As shown, accuracies greater or equal to 90% are achieved for 18 categories, the majority of which show accuracies greater than 95%. An accuracy below 90% is achieved only for the "forest" class, which is easily confused with "mountain" and "river" and thus shows an accuracy of 88%. This result is easily explained by the fact that there are usually trees next to mountains and rivers, making it difficult to distinguish these scenes.

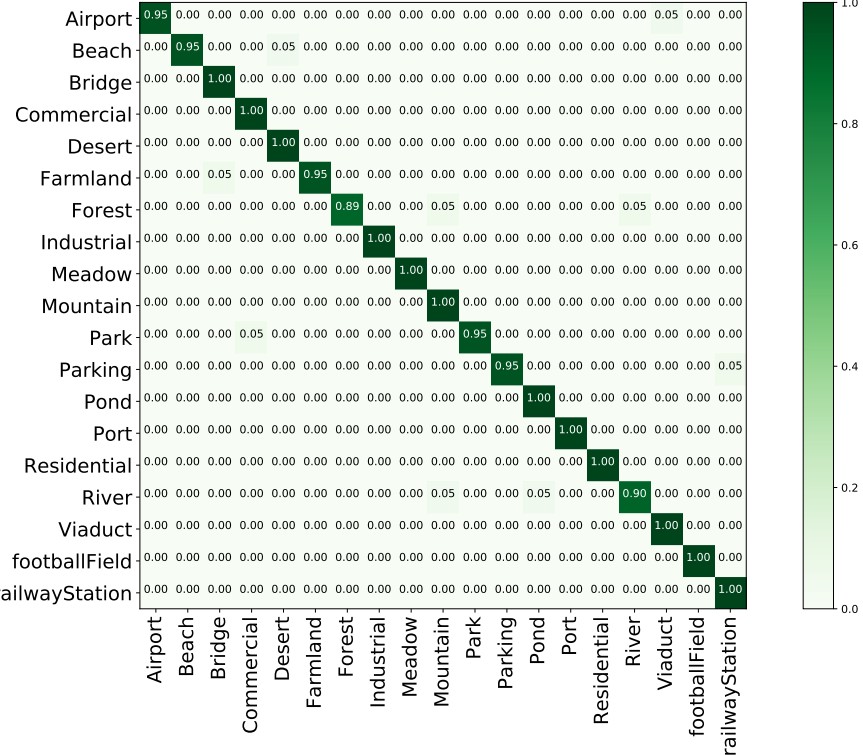

**Figure 13.** Confusion matrix of the proposed method on the WHU-RS19 dataset with a training proportion of 60%.

## 5. Discussion

### 5.1. Result Analysis

Our proposed method achieves accuracies of 94.21%, 96.89%, 99.11%, and 98.98% on the NWPU, AID, UC Merced, and WHU-RS19 datasets, respectively. The networks trained with our proposed

MTL framework significantly outperform all the baselines, demonstrating that our framework can generalize well to various models for remote sensing scene classification. Compared with other methods, our method achieves state-of-the-art results. The following can be seen from the results.

- When trained under our MTL framework, the network models yield significantly improved experimental results without an increase in the number of parameters compared to the baselines.
- Due to the lack of image variations and diversity in the UC Merced and WHU-RS19 datasets, the overall accuracy on these datasets is almost saturated using deep CNN features. By contrast, the NWPU-RESISC45 dataset and AID are more challenging due to their rich image variations, large within-class diversity, and high between-class similarity.
- Compared with the baselines, our framework helps CNN models achieve considerable improvements with little increase in model complexity and training time.
- The proposed MTL framework yields better performance than the baselines when the number of training samples is small. This is because by combining the self-supervised learning and classification tasks, data can be used more effectively.

### 5.2. Parameter Sensitivity

This mixup loss can introduce more randomness into the model, and can improve the feature representation ability of the model. The important parameter $\lambda$ of mixup loss is a random number generated from the $Beta(\alpha, \alpha)$ distribution. The value of $\alpha$ is very important, so we need to evaluate whether this parameter is sensitive to the experimental results.

In order to compare the effects of different $\alpha$ values, we conducted comparative experiments based on ResNet-50 on the four datasets, considering values of $\alpha$ from the set {0.5,1,3}. Figure 14 reports the detailed results. As seen, the different values of $\alpha$ have very little effect on the results, and the different results across the four datasets fluctuate within a very small range. The results suggest that our proposed MTL framework is insensitive to the choice of $\alpha$.

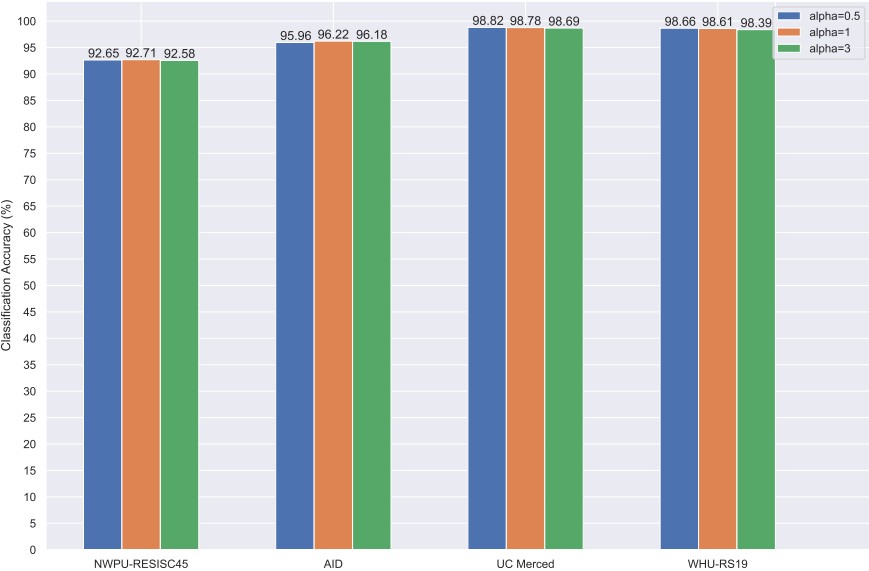

**Figure 14.** Overall accuracy (%) of the proposed method with and without MTL on the four datasets.

### 5.3. Qualitative Analysis and Visualizations

Gradient-weighted class activation mapping (Grad-CAM) [66] is a popular visualization method in which gradients are used to calculate the importance of spatial locations in CNNs. Because the gradients are calculated with respect to a single class, the Grad-CAM results can clearly show attended regions. To visualize whether the networks had learned discriminative features, we applied Grad-CAM to various networks using images from the NWPU-RESISC45 validation set after training with a

training proportion of 20%. By observing the regions that the networks considered important for predicting a class and the confidence scores of the decisions, we attempted to determine which network was able to learn more discriminative features.

Specifically, we compared the confidence scores and visualization results obtained using an MTL-trained network (ResNeXt-101+MTL) with those of the corresponding baseline model (ResNeXt-101). As Figure 15 shows, the model trained using our framework has stronger feature extraction abilities in that it better captures the details that represent semantic features in images with complex backgrounds, and it achieves higher confidence in the classification of some difficult objects than the baseline model does. The visualizations suggest that our MTL framework is capable of removing cluttered backgrounds and gradually focusing on discriminative parts of the remote sensing images.

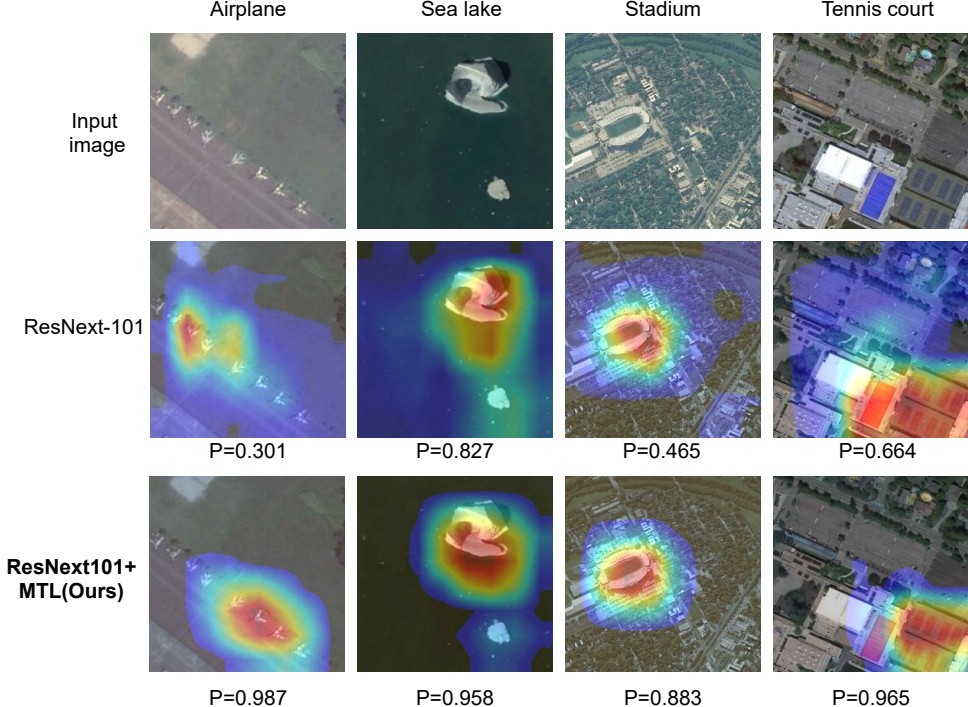

**Figure 15.** Grad-CAM [66] visualization results. We compare the visualization results obtained using an MTL-trained network (ResNeXt-101+MTL) with those of the corresponding baseline model (ResNeXt-101). The Grad-CAM visualization was calculated for the last convolutional outputs. The ground-truth labels are shown above each input image, and P denotes the softmax score of each network for the ground-truth class.

## 6. Conclusions

In this paper, to improve the feature extraction ability of CNN models and allow them to use information from samples more effectively when the sample size is insufficient, we propose an MTL framework that combines the tasks of self-supervised learning and classification. Our proposed MTL framework utilizes the mixup loss strategy to dynamically adjust the weights for MTL, thereby not only improving the classification performance, but also avoiding sensitivity to particular parameter settings.

The proposed MTL framework can help CNN models extract important feature information more effectively and further mitigate the challenges for classification presented by the presence of many small objects and complex backgrounds in images. By introducing image rotation, more image information can be utilized, and more discriminative feature representations can be learned from a

limited amount of data. Our proposed framework can help ResNext-101 achieve accuracies of 94.21%, 96.89%, 99.11%, and 98.98% on the NWPU, AID, UC Merced, and WHU-RS19 datasets, respectively.

Extensive experiments show that features extracted by our multitask learning framework are effective and robust compared with state-of-the-art methods for remote sensing scene classification. Due to the rapid development of self-supervised learning, we have not tried to combine multiple self-supervised learning tasks. In the future work, we will explore more self-supervised learning tasks to further improve the representation ability of network models. We hope that our approach can be applied for other downstream tasks of remote sensing image interpretation.

**Author Contributions:** Z.Z. and Z.L. designed the deep learning model and performed the experiments; Z.Z. and J.L. wrote the paper. C.C. guided the design of the network and checked the experiments; Y.P. reviewed the paper. All authors have read and agreed to the published version of the manuscript.

**Funding:** This work was supported in part by the Strategic Priority Research Program of the Chinese Academy of Sciences(XDA19060205, XDA19020305, and XDA19020104), in part by the Key Research Development Program of China(2019YFC0507405), in part by the National R & D Infrastructure and Facility Development Program of China, "Fundamental Science Data Sharing Platform" (DKA2019-12-02-18), in part by the Special Project of Informatization of Chinese Academy of Sciences (XXH13505-03-205, XXH13506-305, and XXH13506-303).

**Conflicts of Interest:** The authors declare no conflict of interest.

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
