# Peer review of "When Self-Supervised Learning Meets Scene Classification: Remote Sensing Scene Classification Based on a Multitask Learning Framework"

_remotesensing, doi:10.3390/rs12203276_

Round 1

Reviewer 1 Report

The paper presents a new multi-task learning method for remote sensing classification.
The proposed method is based on a mixup loss strategy to combine two tasks with dynamic weighting: image augmentation with progressive 90 degres rotation, and classification.

The paper is well written and detailed. The topic is timely and the proposed method is intriguing and adequately explained.
Competitive results are obtained on four widely adopted remote sensing scene classification datasets.

I particularly liked the part concerning the prediction confidence of the model, using GRAD-CAM.
Overall, this paper presents merit from the methodological and experimental viewpoint.

I have the following suggestions to improve the manuscript:

- Related work is missing approaches purely based on feature extraction, which I suggest adding (see DOI: 10.3390/app10175792, 10.1109/ACCESS.2020.2968771, 10.1109/TGRS.2003.814625)

- The authors propose in addition to the multi-task learning setting, a representation averaging approach based on the different rotations.
However, in the experiments, I could not find a separate result for images using the original representation and the averaged representation.
Was just the averaged representation evaluated? Please clarify.

- Table 3-4-5-6: Calculating a percentage of gain for all the methods with respect to a baseline (GoogLeNet+SVM) would help to emphasize the improvement obtained by the proposed methods.

- Figures 9-10 should be enlarged. Try to match the readability of Figure 12.

I think that after these aspects are addressed the paper will be suitable for publication.

Author Response

Response to Reviewer 1 Comments

Point 1: Related work is missing approaches purely based on feature extraction, which I suggest adding(seeDOI:10.3390/app10175792,10.1109/ACCESS.2020.2968771,10.1109/TGRS.2003.814625).

Response 1: We are grateful for the suggestions. We have added relevant references in the chapter “Introduction”. The corresponding changes are highlighted in the new version.

Point 2: The authors propose in addition to the multi-task learning setting, a representation averaging approach based on the different rotations. However, in the experiments, I could not find a separate result for images using the original representation and the averaged representation. Was just the averaged representation evaluated? Please clarify.

Response 2: Special thanks to you for your good comments. In the new version, we have improved the experimental validation according to the reviewers' suggestions. In section 4.4, we added an experimental comparison of aggregation prediction and single-image prediction. And in the new submission, we combined the document with the point-by-point responses, and the reviewed manuscript with all changes clearly highlighted.

Point 3: Table 3-4-5-6: Calculating a percentage of gain for all the methods with respect to a baseline (GoogLeNet+SVM) would help to emphasize the improvement obtained by the proposed methods.

Response 3: We are incredibly grateful for the suggestions. Considering the comparison with other advanced methods and the size limit of the table, we bolded our proposed methods in table 4-5-6-7 to emphasize the improvement obtained by the proposed methods.

Point 4: Figures 9-10 should be enlarged. Try to match the readability of Figure 12.

Response 4: Special thanks to you for your good comments. In the newly submitted version, we modified the figures 9-10 to increase the readability.

Reviewer 2 Report

The issue addressed in this paper is of interest to scientists working on the development and application of convolutional neural networks (CNNs), and in particular in the exploitation of images in remote sensing. In this paper, the authors propose a methodology, based on a multitasking learning framework, allowing the improvement of the extraction of the characteristics of the images and also a better exploitation of the extracted knowledge. The different simulations carried out using four datasets, and the results presented are encouraging.
However, reading the content of the submitted paper, it is very difficult to appreciate the contribution of the paper. Consequently, it would be useful to reinforce the majority of the sections presented, and to take into account the remarks and comments given below.
- It would be useful to reinforce the motivations and interest aspects of the methodology developed and in particular in a practical and / or operational context. The methodology adopted should be presented, for example in the introduction, in the form of a block diagram with inputs and outputs for each of the steps and / or tasks.
- It would be important to analyze the influence of the different bricks making up the overall treatment process, starting with the quality of the images, going through the preprocessings used, the influence of the speckle, the resolution of the images, and the results obtained quantified.
- Taking into account the previous point, it would be useful to indicate the limits of the methodology adopted, the algorithmic complexity of the various algorithms, and in particular the computation time necessary to perform the various simulations presented.
- It would be useful to specify how to implement the methodology developed in a practical and / or operational context.
- It would be useful to review the quality of certain figures.
- As indicated at the beginning, it would be useful to reinforce the conclusion with a presentation of the scientific advances, the contribution and the limits of the methodology adopted. The prospects are to be further developed.

Author Response

Response to Reviewer 2 Comments

Point 1: It would be useful to reinforce the motivations and interest aspects of the methodology developed and in particular in a practical and / or operational context. The methodology adopted should be presented, for example in the introduction, in the form of a block diagram with inputs and outputs for each of the steps and / or tasks.

Response 1: Special thanks to you for your good comments. It is really true as reviewer suggested that we should reinforce the motivations and present the methodology we adopted. Considering the reviewer's suggestion, we reinforced the motivation and added a flowchart of the proposed framework in the section of introduction. The corresponding changes are highlighted in the new version.

Point 2: It would be important to analyze the influence of the different bricks making up the overall treatment process, starting with the quality of the images, going through the preprocessing used, the influence of the speckle, the resolution of the images, and the results obtained quantified.

Response 2: We are incredibly grateful for the suggestions. In the new version, we have improved the experimental validation according to the reviewers' suggestions. We added a flowchart of the proposed methods in the section of introduction to specify the overall process. In section 4.4, we added an experimental comparison of aggregation prediction and single-image prediction. We also added the related contents in the conclusion section.

Point 3: Taking into account the previous point, it would be useful to indicate the limits of the methodology adopted, the algorithmic complexity of the various algorithms, and in particular the computation time necessary to perform the various simulations presented.

Response 3: We are deeply sorry for our negligence of these parts. We have re-written these parts to the reviewer's suggestion. Our proposed framework does not change the original CNN network structure. It does not introduce new parameters. We also added discussion about computation time and model's complexity in section “Result Analysis”.

Point 4: It would be useful to specify how to implement the methodology developed in a practical and / or operational context.

Response 4: Special thanks to you for your good comments. We added a flowchart of the proposed framework in the section of introduction to specify how to implement the methodology. We also made relevant supplements in the section 4.2 named implementation details.

Point 5: It would be useful to review the quality of certain figures.

Response 5: We are incredibly grateful for the suggestions. In the newly submitted version, we modified the figures 9-10 to increase the readability and quality.

Point 6: As indicated at the beginning, it would be useful to reinforce the conclusion with a presentation of the scientific advances, the contribution and the limits of the methodology adopted. The prospects are to be further developed.

Response 6: Many thanks for the valuable comments and suggestions. We have re-written this part to the reviewer's suggestion. Extensive experiments show that features extracted by our multitask learning framework are effective and robust compared with state-of-the-art methods for remote sensing scene classification. Due to the rapid development of self-supervised learning, we have not tried to combine multiple self-supervised learning tasks. In the future work, we will explore more self-supervised learning tasks to further improve the representation ability of network models. We hope that our approach can be applied for other downstream tasks of remote sensing image interpretation.

Round 2

Reviewer 1 Report

The authors addressed all my concerns and the manuscript has improved significantly. I suggest it be accepted for publication.

This manuscript is a resubmission of an earlier submission. The following is a list of the peer review reports and author responses from that submission.